# Two-Year Reproducibility of Axial Length Measurements after Combined Phacovitrectomy for Epiretinal Membrane, and Refractive Outcomes

**DOI:** 10.3390/jcm9113493

**Published:** 2020-10-29

**Authors:** Tae Seen Kang, Yong-Il Shin, Cheon Kuk Ryu, Jung Yeul Kim

**Affiliations:** 1Department of Ophthalmology, Chungnam National University Hospital, Daejeon 35015, Korea; tskang85@naver.com (T.S.K.); 2mail01@hanmail.net (Y.-I.S.); chkryu6@gmail.com (C.K.R.); 2Department of Ophthalmology, Gyeongsang National University Changwon Hospital, Changwon 51472, Korea

**Keywords:** axial length, combined phacovitrectomy, reproducibility, postoperative prediction errors

## Abstract

Purpose: To determine the long-term reproducibility of axial length measurements and mean postoperative prediction errors after combined phacovitrectomy in patients with idiopathic epiretinal membranes. Design: Retrospective cohort study. Methods: The study included 43 patients who underwent combined phacovitrectomy and 30 patients who underwent only phacoemulsification. To determine the effect of vitrectomy, we compared patients treated with phacoemulsification only versus those treated with combined phacovitrectomy. Axial lengths were measured three times with a one-year interval, and the intraclass correlation coefficient (ICC), coefficient of variation (CV), and test–retest standard deviation (TRTSD) were assessed. Results: There was no significant change in axial length, and axial length measurements showed high reproducibility in all groups. ICC, CV, and TRTSD values were 0.997, 0.24%, and 0.056, respectively, for the vitrectomized eyes. The mean postoperative prediction error was −0.37 diopters(D) in vitrectomized eyes (*p* < 0.001), while it was +0.11 D in patients with phacoemulsification (*p* = 0.531). The myopic shift was more obvious in eyes with a shallower anterior chamber (*p* = 0.008) and a thicker lens (*p* = 0.025). Conclusions: Axial length measurements showed excellent long-term reproducibility at 2 years after combined phacovitrectomy. Myopic shifts were observed after combined phacovitrectomy, which was probably due to changes in the effective lens position after combined phacovitrectomy, rather than to changes in the axial length.

## 1. Introduction

The epiretinal membrane (ERM), first described by Iwanoff in 1865, has been known by many names, including primary retinal folds, macular pucker, preretinal macular fibrosis, surface wrinkling retinopathy, preretinal macular fibrosis, internal retinal fibrosis, and cellophane maculopathy [1]. It appears as a glossy thin film on the macula resulting from proliferation of glial cells in front of the internal limiting membrane (ILM). It occurs in patients ≥50 years of age and in 6% of the total population [2]. Since the 1980s, vitrectomy and ERM removal have been conducted, and excellent results have been reported in long-term follow-ups [3].

The surgical outcomes have improved with several techniques, including ILM peeling with indocyanine green dye [4] and the use of triamcinolone as a vital dye [5]. New surgical methods such as nonvitrectomizing vitreous surgery, a surgical procedure aiming to remove ERM without removing the vitreous, are being developed to minimize cataract [6]. However, vitrectomy and ERM removal are still standard treatments. Thus, in most ERM surgeries, the surgeon should consider cataract surgery, because most patients are elderly and the vitrectomy might induce a cataract [7]. Nowadays, combined phacovitrectomy, which consists of pars plana vitrectomy, phacoemulsification, and intraocular lens (IOL) implantation, is preferred because it reduces the number of surgeries, clarifies retinal anatomy by removing lens opacity, and reduces the inconvenience to patients [8]. At this time, the surgical goal includes not only ERM removal but also correction of the patient’s refractive error. However, previous studies have reported that combined phacovitrectomy for ERM can result in myopic shift [9,10,11]. This myopic shift reduces patient satisfaction after combined phacovitrectomy and makes it difficult to determine the IOL power.

The purpose of this study was therefore to determine the long-term postoperative prediction errors after combined phacovitrectomy. Because a previous report stated that axial lengths increased after ERM surgery [12], we also measured axial lengths over the same time period to determine the cause of the postoperative prediction errors.

## 2. Methods

### 2.1. Study Design and Participants

We retrospectively analyzed medical records, and the study protocol was approved by the institutional review board of Chungnam National University Hospital. This study was conducted in accordance with the tenets of the Declaration of Helsinki. The requirement for obtaining informed patient consent was waived due to the retrospective nature of the study. Patients with ERM and patients with cataract who were admitted to Chungnam University Hospital between January 2011 and June 2016 were included in this study. Exclusion criteria were treatment with silicone oil or gas tamponade during surgery, a history of intraocular surgery, retinal disorders besides ERM, follow-up cessation within 2 years of treatment, and media opacity sufficient to disrupt axial length measurements.

Vitrectomized eyes of patients fulfilling the inclusion criteria were included in the study group. The fellow eyes were included in the control group as long as they did not correspond to the exclusion criteria. We also tried to evaluate the effect of vitrectomy by comparing the eyes of the phacoemulsification-only group with those of the control group during the same period.

During the follow-up period, each participant underwent refraction measurements using an automatic refractometer, measurements of best-corrected visual acuity (BCVA), slit-lamp biomicroscopy, a fundus examination, an optical coherence tomography scan, an intraocular pressure test with a non-contact tonometer, and an axial-length measurement.

### 2.2. Axial Length Measurements and IOL Power Calculations

Axial lengths were measured using partial interferometry (IOL Master^®^, Carl Zeiss, Jena, Germany) before surgery in both eyes. The axial length was measured by asking patients to fixate at the system’s fixation target. The axial length was then automatically calculated, and it was defined as the mean value of at least 10 valid measurements. During the follow-up, the axial length was measured twice at intervals of >1 year.

We calculated the IOL power using the SRK-T formula. The mean postoperative prediction error was defined as the difference between the expected refraction before surgery and the spherical equivalent measured at 2 years after surgery.

### 2.3. Surgical Methods

Combined 23-gauge sutureless phacovitrectomies were performed under retrobulbar anesthesia by a single surgeon (J.Y.K.). Cataract extraction preceded pars plana vitrectomy after the insertion of a trocar. A 2.8 mm clear corneal incision was made using a superior approach, and standard phacoemulsification was performed. Vitrectomy was performed to remove the ERM after the anterior chamber had been filled with a viscoelastic substance. The Constellation Vision System^®^ (Alcon, Fort Worth, TX, USA) was used for vitrectomies. After removal of the ERM using 23G intraocular forceps, the ILM was stained with indocyanine green dye (Diagnogreen; Daiichi Sankyo, Tokyo, Japan). It was then removed with end-grip forceps. A three-piece spherical acrylic IOL with a 6.0 mm optical zone diameter (Sensar AR40e^®^; Abbot Medical Optics, Santa Ana, CA, USA) was inserted into the capsular bag. An incision suture was not made.

In the phacoemulsification group, cataract surgery was performed using the same method as with combined phacovitrectomy. We used the same IOL power calculation formula for myopia closest to emmetropia and inserted the same IOL in the combined phacovitrectomy group.

### 2.4. Statistical Analyses

SPSS statistical software for Windows, version 24.0 (IBM Corp., Armonk, NY, USA) was used for all statistical analyses. Normality tests were performed via the Shapiro–Wilcoxon test in SPSS. We conducted independent Student’s *t*-tests to compare two variables and repeated-measures ANOVA for comparing three variables. The *p*-values of the multiple tests were adjusted with the Bonferroni method. A comparative analysis of qualitative variables was performed using the chi-squared test. The long-term reproducibility was determined using the intraclass correlation coefficient (ICC), coefficient of variation (CV), and test–retest standard deviation (TRTSD). The TRTSD was calculated as follows: axial lengths were measured three times over 2 years and averaged, and the standard deviation of the mean value of each measured value was defined as the TRTSD. The ICC was calculated by dividing the within-subject variance by the total variance, and the CV was calculated by dividing the TRTSD by the mean of the total measured values, then multiplying by 100. A value of *p* < 0.05 was considered statistically significant.

For correlation analysis, keratometry, axial length, and anterior chamber depth (ACD) were measured using partial interferometry, and lens thickness was measured using ultrasound.

### 2.5. Data Availability

Data supporting the findings of the current study are available from the corresponding author upon a reasonable request.

## 3. Results

### 3.1. Participants

Between January 2011 and June 2016, 345 patients were diagnosed with ERM, of which 223 were treated with combined phacovitrectomy. Of these patients, 43 (11 males and 32 females with a mean age of 64.0 ± 8.3 years) were followed up for >2 years after surgery.

During the same period, 119 patients were diagnosed with cataract only and underwent only phacoemulsification, including 30 patients (12 males and 18 females; mean age, 67.0 ± 12.2 years) who were followed up for 2 years. None of the patients developed complications requiring additional surgery within 2 years.

### 3.2. Visual Acuity and Central Macular Thickness (CMT)

At the first examination, the mean best-corrected visual acuity (BCVA) of vitrectomized eyes using the logMAR chart was 0.19 ± 0.22. One year and 2 years after surgery, the mean BCVA values of vitrectomized eyes were 0.06 ± 0.09 and 0.05 ± 0.07. Analysis of variance (ANOVA) showed significant improvement in BCVA values for vitrectomized eyes (Table 1).

In vitrectomized eyes, the mean CMT measured by optical coherence tomography was 426 ± 91 μm before surgery. At 1 and 2 years after vitrectomy and ERM removal, the mean CMTs were 375 ± 44 and 360 ± 37 μm, respectively. There was a significant decrease in the CMT of 66 μm before and after surgery (*p* < 0.001); additionally, a more significant reduction was associated with a thicker CMT before surgery (*p* = 0.003).

### 3.3. Reproducibility of Axial Length Measurements and Keratometry Using Partial Interferometry

The mean preoperative axial length of vitrectomized eyes was 23.77 ± 0.96 mm. It was 23.73 ± 0.96 mm at 1 year and 23.75 ± 0.97 mm at 2 years after surgery. The mean axial lengths of the fellow eyes were 23.65 ± 1.03 mm before surgery, 23.69 ± 1.05 mm at 1 year after surgery, and 23.67 ± 1.04 mm at 2 years after surgery. The mean preoperative axial length of eyes treated with phacoemulsification was 23.60 ± 1.29 mm, and it was 23.50 ± 1.26 and 23.49 ± 1.24 mm at 1 and 2 years, respectively, after surgery. The ANOVA results revealed that axial lengths were consistent over the 2 years (Table 1). We compared initial and final axial lengths of each of the three groups using scatterplots, which showed that axial lengths measured 2 years after surgery did not differ from those measured before surgery in all three groups (Figure 1).

We analyzed the confidence of axial length measurements using ICC, CV, and TRTSD. High reproducibility was shown in all three groups. The ICC, CV, and TRTSD values were 0.997, 0.24%, and 0.056, respectively, for the vitrectomized-eye group; 0.999, 0.26%, and 0.061, respectively, for the phacoemulsification group; and 0.999, 0.23%, and 0.054, respectively, for the fellow-eye group (Table 2).

Keratometry values as measured by partial interferometry did not change significantly in either the vitrectomized or fellow eyes over 2 years.

### 3.4. Postoperative Prediction Errors

In the phacovitrectomy group, the mean preoperative refraction was −0.03 ± 1.93 D, and predicted refraction through the SRK-T formula was −0.53 ± 0.33 D. One year after phacovitrectomy, the mean measured refraction was −0.85 ± 0.65 D, and it was −0.90 ± 0.55 D 2 years after surgery (ANOVA; *p* = 0.003). There was a significant difference between the predicted refraction and measured refraction 1 year and 2 years after surgery in the Bonferroni test (*p* = 0.02, *p* = 0.005, respectively). The mean postoperative prediction error after 2 years was −0.37 ± 0.48 D (*p* < 0.001), which indicated a significant myopic shift.

In the phacoemulsification group, predicted refraction was −0.77 ± 0.50 D. One year after phacoemulsification, the mean refraction was −0.92 ± 0.78 D, and it was −0.66 ± 0.97 D 2 years after surgery (ANOVA; *p* = 0.372). The mean postoperative prediction error at 2 years after phacoemulsification indicated the presence of slight hyperopia, but the error was not significant (0.11 ± 0.90 D; *p* = 0.531).

To determine the causes of postoperative prediction error, we performed correlation analyses of the axial length, preoperative ACD, differences in postoperative and preoperative ACD, preoperative lens thickness, and preoperative CMT. A myopic shift was apparent when the preoperative ACD was shallower (*p* = 0.008), and the preoperative lens was thicker (*p* = 0.025). The postoperative decreases in CMT (*p* = 0.275) and preoperative CMT (*p* = 0.939) were not significantly correlated with the myopic shift. The axial length was correlated with postoperative prediction error, but not significantly (*p* = 0.112) (Figure 2). There was no significant correlation between the postoperative prediction error and preoperative axial length, ACD, or lens thickness in the phacoemulsification group. Eyes that underwent combined phacovitrectomy exhibited a significant myopic shift compared with eyes that underwent phacoemulsification (*p* = 0.006).

### 3.5. Changes in Preoperative and Postoperative Anterior Chamber Depth

The mean ACD of vitrectomized eyes was 3.29 ± 0.39 mm. It was 4.40 ± 0.55 mm at 2 years after surgery. In the phacoemulsification group, the mean ACD was 3.07 ± 0.40 mm before surgery and 4.60 ± 0.62 mm after 2 years. ACD change in the phacovitrectomy group was 1.10 mm, while that of the phacoemulsification group was 1.53 mm (Table 3, *t*-test; *p* = 0.010).

### 3.6. Univariate and Multivariate Linear Regression Analyses for Postoperative Prediction Error

Univariate and multivariate linear regression analyses were performed to determine the clinical factors that cause postoperative prediction errors in the phacovitrectomy and phacoemulsification group (Table 4). Univariate linear regression analysis indicated that a type of surgery (*p* = 0.003) and changes in ACD (*p* = 0.014) was associated with the postoperative prediction error. The other biometric variables, axial length and preoperative ACD, displayed no significant association. In the multivariate regression analysis, only the type of surgery was associated with the myopic shift (*p* = 0.026).

## 4. Discussion

In previous studies, the mean postoperative prediction error after phacoemulsification was approximately zero [13]. Although the mean postoperative refraction was very close to the predicted value, each patient retained some error after surgery. The errors after IOL implantation result from various causes. Past methods of axial length measurements were conducted using ultrasound and were significantly influenced by the patient’s cooperation and the skill of the examiner. A study with ultrasound showed that axial length measurement errors accounted for 54% of all postoperative prediction errors [14]. After the introduction of partial interferometry, the error rate decreased to 17% [15] because partial interferometry is more reliable than ultrasound and less affected by the skill of the examiner [16]. There have been several reports that the axial length measured with ultrasound increases significantly as the macular edema decreases after combined phacovitrectomy for ERM [12,17]. However, we measured axial length using partial interferometry as it is less affected by macular edema [18].

Although outcomes may vary according to the literature, an axial-length measurement error of 0.1 mm may cause an error of 0.28 D [14]. In the present study, the CMTs decreased by as much as 66 μm. If the decrease in macular edema had affected axial-length measurement, the axial length might have increased by approximately 0.07 mm, and the final refraction would have shifted to myopia with a value of 0.2 D. However, because the axial lengths in the present study were unchanged, the myopic shift in this study did not result from CMT decreases, and we again confirmed that partial interferometry was less affected by macular edema.

Myopic shifts after combined phacovitrectomy have been reported in several studies, and these shifts were independent of underlying retinal diseases and biometry methods. Gas tamponade for treating macular holes or retinal detachment can result in a myopic shift [10,19,20,21,22,23], and the use of long-lasting gas results in more significant changes [24]. In addition, postoperative myopic errors have also been reported when combined phacovitrectomy was performed in patients with ERM or macular edema [9,10,11,12,17,19,22,23,24,25,26,27]. In our previous study, we also found that a myopic shift of −0.41 D occurred after combined phacovitrectomy in macular-sparing rhegmatogenous retinal detachment [28].

This myopic shift could occur when the axial length is lengthened or the cornea steepened. Even though many studies reported consistent changes in myopia, there were various hypotheses explaining these changes. Kovacs [12] reported that the mean axial length increased after ERM surgery. Conversely, Falkner [10] reported that combined phacovitrectomy shortened the axial length more than phacoemulsification did, and it resulted in a myopic shift. There were also reports that the cornea became steeper after surgery [10,29]. Hwang [20] reported that the error varied depending on the type of IOL; however, another study reported that the types of IOL had no effect [10]. However, because these studies only compared the results up to 12 months postoperatively, the long-term result was not reported. In the present study, we determined long-term postoperative prediction errors (2 years). Moreover, we investigated whether the changes were due to changes in biometrics such as axial length.

In the present study, the axial length and keratometry values did not change after surgery, and both showed high reproducibility. This is consistent with our previous study [28]. It is also consistent with a previous report stating that partial interferometry is a highly reliable and reproducible test [16]. Because high reproducibility was consistently demonstrated in the fellow-eye (without vitrectomy), combined phacovitrectomy, and phacoemulsification groups, reproducibility rate was not affected by whether patients underwent surgery or not. Thus, the changes in myopia after combined phacovitrectomy for ERM did not occur as a result of changes in axial length or changes in keratometry. Therefore, the cause of the myopic shift after combined phacovitrectomy was an effective lens position (ELP) shift. Because refraction of the pseudophakic eye occurs in the cornea and the IOL, the distance from the corneal apex to the IOL affects the focal length. The distance between the cornea apex and the IOL is called the ELP (Figure 3A), and a 0.25 mm shorter ELP causes myopia of 0.5 D in small eyes [30].

There is no way to measure the position of an IOL in the capsular bag accurately before phacoemulsification. However, according to Olsen’s c-constant, IOL position is affected by the ACD and lens thickness (Figure 3B), while the axial length and corneal curvature have little effect on IOL position [31]. In the present study, correlation analyses between postoperative prediction errors and preoperative factors showed that eyes with thicker lenses and shallower ACDs were associated with significant postoperative prediction errors resulting in myopia after combined phacovitrectomy.

The postoperative prediction error after phacoemulsification was not significant. Based on these observations, the lens position in eyes after combined phacovitrectomy differs from that after phacoemulsification. Performing combined phacovitrectomy results in a shallower ELP and myopic shift than performing cataract surgery alone (Figure 3C).

There is controversy in how ACD changes after combined phacovitrectomy. There are reports that the ACD has deepened [10] or shallowed [32,33]. In this study, changes in ACD were significantly smaller in the phacovitrectomy group than the phacoemulsification group. Besides, in univariate analysis conducted with the phacovitrectomy and phacoemulsification group, there was a significant correlation between ACD change and postoperative prediction error. A possible explanation is that vitreous and posterior lens capsule interactions disappear after vitrectomy, resulting in a shift in the IOL. In the human eye, the vitreous and lens are connected via Weigert’s ligament and Cloquet’s canal. Therefore, when a patient is treated with combined phacovitrectomy, we propose that the procedure affects the posterior surface of the lens during wound healing and capsular fibrosis around the IOL, which may affect the position of the IOL. As such, the target refraction will be achieved by using an IOL which has a lower diopter than the calculated power.

Limitations of this study include the small number of patients and the retrospective design of the study. Unfortunately, due to the lack of longitudinal postoperative data (refractive error, axial length, etc.), we could not provide a more specific evidence for the results of this study. Prospective research enrolling a larger number of patients will be needed to confirm the findings of this study. Second, ACD change did not show a significant correlation in multivariate regression. Third, decreased macular edema could have affected the IOL power calculation and refraction measurements. However, partial interferometry is less affected by macular edema, and axial lengths were unchanged when measured with partial interferometry over 2 years. Several studies indicated that PCI has a limitation to measure ACDs compared to measure axial length [16,34], but we thought it was insignificant to affect the results. Furthermore, the decrease in macular edema and the myopic shift were not significantly correlated; thus, a reduction in macular edema might have had little or no effect on the IOL power calculation.

The advantage of this study is that it measured the axial lengths of patients over a long period and compared the results of patients treated with combined phacovitrectomy and phacoemulsification alone. In patients treated with only phacoemulsification, there was no difference in the predicted and final refractions; however, patients treated with combined phacovitrectomy showed a significant myopic shift, so we should consider this change when determining IOL power.

In conclusion, the axial lengths were not altered within 2 years after combined phacovitrectomy or phacoemulsification. The reproducibility of axial length measurements was high in eyes treated with combined phacovitrectomy or phacoemulsification. Two years after surgery, a significant myopic shift was found in the eyes treated with combined phacovitrectomy compared with those treated with phacoemulsification, and this postoperative prediction error was significantly correlated with shallower ACD and a thicker lens. This change probably resulted from changes in the IOL position in the absence of the vitreous after vitrectomy, rather than from changes in the axial length.

## Figures and Tables

**Figure 1 jcm-09-03493-f001:**
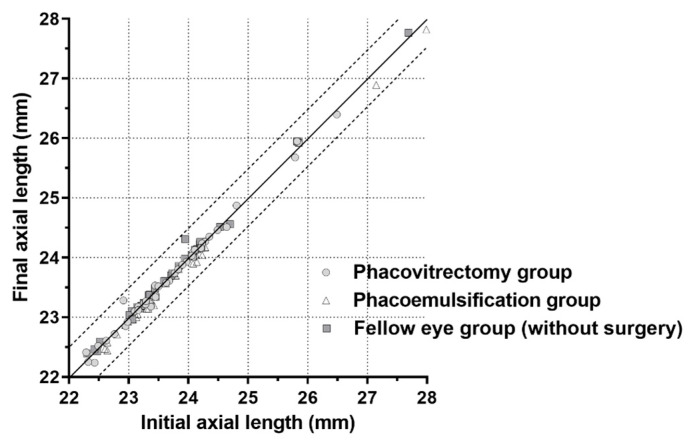
Scatterplot showing the initial and final axial lengths in all groups. The circles, triangles, and squares between the two lines show axial lengths that changed ≤0.5 mm within 2 years.

**Figure 2 jcm-09-03493-f002:**
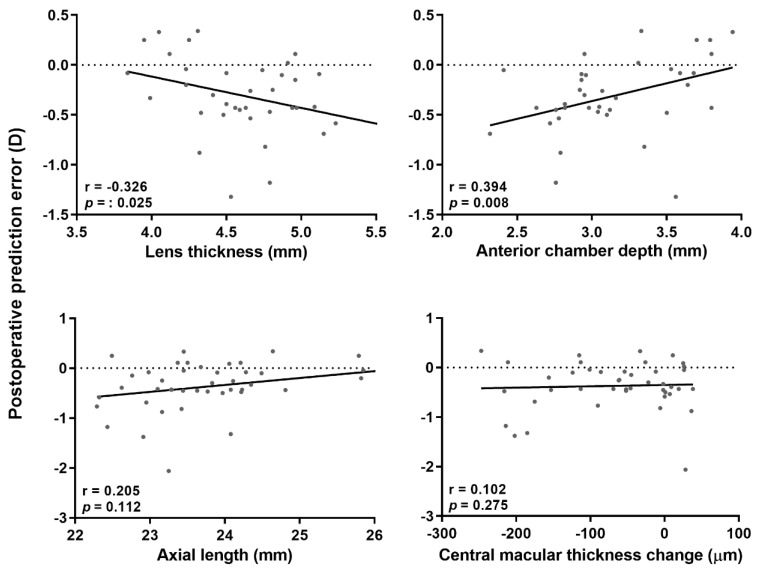
Correlation analysis of the relationships between preoperative factors and postoperative prediction errors (observed refraction – predicted refraction) for vitrectomized eyes. A relatively shallow anterior chamber and a thicker lens resulted in a significant myopic shift. The equations fitted to the regression lines for the regression analyses were as follows: y = −0.314x + 1.144 (*p* = 0.049, top left), y = 0.357x − 1.429 (*p* = 0.016, top right), y = 0.139x − 3.663 (*p* = 0.072, bottom left), and y = 0.000x − 0.349 (*p* = 0.771, bottom right), respectively.

**Figure 3 jcm-09-03493-f003:**
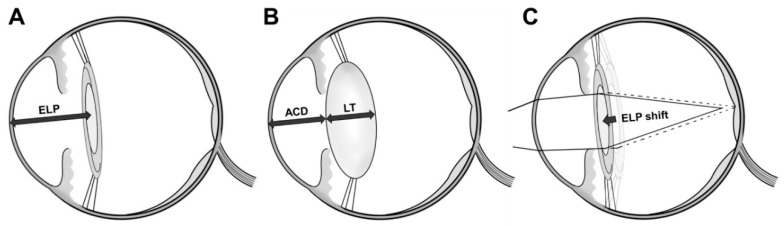
Schematic diagram of changes in postoperative effective lens position (ELP). (**A**): The distance from the corneal apex to the intraocular lens is called the ELP. (**B**): The ELP is affected by anterior chamber depth (ACD) and lens thickness (LT), and the ELP is longer than the ACD and shorter than ACD + LT. (**C**): A shortened ELP results in a shortened focal length and causes a myopic shift.

**Table 1 jcm-09-03493-t001:** Changes in biometrics over 2 years in the phacovitrectomy, phacoemulsification, and fellow-eye groups.

	Baseline	Postoperative 1 Year	Postoperative 2 Years	*p*-Value *
**Phacovitrectomy group**				
BCVA (logMAR)	0.19 ± 0.22	0.06 ± 0.09	0.05 ± 0.07	<0.001
Refraction (D)	−0.03 ± 1.93	−0.85 ± 0.65	−0.90 ± 0.55	0.002
Keratometry (D)	44.13 ± 1.46	44.23 ± 1.43	44.34 ± 1.50	0.804
Axial length (mm)	23.77 ± 0.96	23.73 ± 0.96	23.75 ± 0.97	0.987
**Phacoemulsification group**				
BCVA (logMAR)	0.54 ± 0.48	0.20 ± 0.39	0.16 ± 0.34	0.001
Refraction (D)	−2.12 ± 4.77	−0.92 ± 0.77	−0.66 ± 0.97	0.150
Keratometry (D)	44.34 ± 1.24	44.42 ± 1.26	44.38 ± 1.27	0.972
Axial length (mm)	23.60 ± 1.29	23.50 ± 1.26	23.49 ± 1.24	0.944
**Fellow-eye group (without surgery)**				
BCVA (logMAR)	0.11 ± 0.28	0.11 ± 0.27	0.08 ± 0.24	0.888
Refraction (D)	−0.44 ± 2.01	−0.53 ± 2.26	−0.55 ± 2.34	0.985
Keratometry (D)	44.45 ± 1.07	44.41 ± 1.01	44.50 ± 1.03	0.954
Axial length (mm)	23.65 ± 1.03	23.69 ± 1.05	23.67 ± 1.04	0.987

BCVA, best-corrected visual acuity; D, diopters; logMAR, logarithm of the minimum angle of resolution. Values are expressed as the means ± standard deviations.* From analysis of variance.

**Table 2 jcm-09-03493-t002:** Long-term reproducibility of axial length measurements in the phacovitrectomy, phacoemulsification, and fellow-eye groups.

	ICC	CV (%) *	TRTSD †
Phacovitrectomy group	0.997	0.24	0.056
Phacoemulsification group	0.999	0.26	0.061
Fellow-eye group (without surgery)	0.999	0.23	0.054

ICC, intraclass correlation; CV, coefficient of variation; TRTSD, test–retest standard deviation. * Calculated as 100× within-subject standard deviation/average of the measurements. † Calculated as the square root of the within-subject variance.

**Table 3 jcm-09-03493-t003:** Comparison of changes in ocular biometry between the phacoemulsification group and the phacovitrectomy group.

	Phacovitrectomy Group	Phacoemulsification Group	*p*-Value
Postoperative prediction error *	−0.37 ± 0.48	0.11 ± 0.90	**0.006**
Changes in anterior chamber depth †	1.10 ± 0.66	1.53 ± 0.67	**0.010**

* observed refraction (2-years)—predicted refraction; † postoperative anterior chamber depth (2-years)—preoperative anterior chamber depth. Boldface numbers indicate statistically significant differences at *p* < 0.05.

**Table 4 jcm-09-03493-t004:** Univariate and multivariate linear regression analyses between various clinical factors and postoperative prediction error.

	Univariate Regression		Multivariate Regression	
	β ± SE	*p*-Value	β ± SE	*p*-Value
Surgery (0 = phacoemulsification, 1 = phacovitrectomy)	−0.296 ± 0.096	**0.003**	−0.228 ± 0.100	**0.026**
Axial length	0.026 ± 0.047	0.581		
ACD (Preoperative)	−0.014 ± 0.125	0.909		
Preoperative lens thickness	−0.021 ± 0.109	0.847		
Changes in ACD *	0.187 ± 0.074	**0.014**	0.138 ± 0.074	0.070

ACD = anterior chamber depth; postoperative prediction error = observed refraction (2 years)—predicted refraction; * postoperative ACD (2 years)—preoperative ACD; boldface numbers indicate statistically significant differences at *p* < 0.05.

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
