# Peer review of "Two-Year Reproducibility of Axial Length Measurements after Combined Phacovitrectomy for Epiretinal Membrane, and Refractive Outcomes"

_jcm, 2020, doi:10.3390/jcm9113493_

Round 1
Reviewer 1 Report
Please refer attached file.

Author Response
Thanks for a very worthy opinion.
We wrote replies in a separate word file.

Reviewer 2 Report
I have no additional comments.
Author Response
Thanks for your attentive comment.
Reviewer 3 Report
Did all variables show a normal distribution? What kind of test was used?
Results in Table 1 are based on an ANOVA, which has not been reported in the methods section. Did the authors adjust p-values for multiple testing? If yes, how did they do this?
Is there an equation reported for the regression analysis?
Rarely there is a typo, which needs to be corrected, but otherwise well done.
Author Response
Did all variables show a normal distribution?
Thank you for your comment. We performed the normality test via the Shapiro-Wilcoxon test in SPSS. The continuous variables were satisfied with the normality test with p-values over 0.05. We added it to the method section. (Line 100)
Results in Table 1 are based on an ANOVA, which has not been reported in the methods section. Did the authors adjust p-values for multiple testing? If yes, how did they do this?
Thank you for your constructive advice. When comparing two variables, we performed the independent Student's t-test, and when comparing three variables, we performed the repeated-measures ANOVA. We adjusted the p-values of the multiple tests with the Bonferroni method. We added it to the method section. (Line 101)
Is there an equation reported for the regression analysis?
Thank you for your comment. We added the equations in the legend of Figure 2. In addition, it was thought that additional equation description in Table 4 would not be necessary, so we did not add. Once again, thank you for your kind advice.
Rarely there is a typo, which needs to be corrected, but otherwise well done.
Thanks for your attentive comment.
This manuscript is a resubmission of an earlier submission. The following is a list of the peer review reports and author responses from that submission.
Round 1
Reviewer 1 Report
Authors found that a significant myopic shift was found in the eyes treated with combined phacovitrectomy compared with those treated with phacoemulsification, and this postoperative prediction error was significantly correlated with shallower ACD and a thicker lens.
This change probably resulted from changes in the IOL position in the absence of the vitreous after vitrectomy, rather than from changes in the axial length.
The manuscript is well written. Limitations and advantages of the study are well focused.
Only a minor issue:
In the introduction could be usefull to add about the possible treatment options for ERM the possibility described by Reibaldi et al.(Retina 2015) of a transconjunctival nonvitrectomizing vitreous surgery (NVS) in case of ERM.
This technique is an effective surgical procedure in eyes with epiretinal membrane and it induces less progression of nuclear sclerosis than 25-gauge vitrectomy.
Author Response
Q) In the introduction could be useful to add about the possible treatment options for ERM the possibility described by Reibaldi et al.(Retina 2015) of a transconjunctival nonvitrectomizing vitreous surgery (NVS) in case of ERM.
Thanks for the review. As you said, we have added a reference about transconjunctival convitrectomizing vitreous surgery to the introduction(line 40~43)

Reviewer 2 Report
Dr. Kang et al. have submitted their original manuscript, entitled “Long-term reproducibility of axial length measurements after combined phacovitrectomy for epiretinal membrane, and refractive outcomes” to Journal of Clinical Medicine.
My major concerns are as follows:
- The purpose of this study was to determine the long-term postoperative prediction errors after combined phacovitrectomy. Why did the authors need to investigate “long-term” postoperative prediction errors? It has been demonstrated that refraction is stable 3 months after cataract surgery (Hoffer KJ, et al. Am J Ophthalmol 2015;160:403-5.e1.). If the authors wanted to investigate postoperative prediction errors after phacovitrectomy, they could evaluate prediction errors at an earlier period, not 2 years after surgery, which would led to enroll more patients. Although the authors mentioned that postoperative prediction errors were correlated with lens thickness (p = 0.041) and ACD (p = 0.049), and not with AL (p = 0.051), there is little difference in p values between these 3 parameters. The authors should analyze the data with larger number of patients.
- If the authors insist that the lens position in eyes after phacovitrectomy differs from that after phacoemulsification, they should provide the data of preoperative and postoperative ACD in both groups.
- It has been well known that the PCI AL is not influenced by retinal edema, in contrast to that measured by the ultrasound (ref. 10). Therefore, the authors should not equate papers in which ALs were measured by ultrasound to those that used PCI. In addition, they did not need to discuss the effect of macular edema on IOL power calculation by PCI. Moreover, the authors should not discuss the relationship between AL measurement by ultrasound and refractive errors because they used PCI in this study and everyone knows that the two systems have different characteristics.
- The authors should perform multivariable analysis to investigate the correlation between postoperative prediction error and preoperative ocular parameters because there is an interaction between preoperative parameters (for example, thicker lens leads to shallower ACD and shorter AL, as the authors mentioned).
My minor concerns are listed below:
- Introduction is too redundant, especially 1st paragraph.
- P2, line78-9: Why did the authors measure ALs with not only PCI but also ultrasaound? Because, as the authors mentioned, PCI is more reliable than ultrasound.
- P4, line 147: AL valued are different from those in Table 1.
- P7, line214: What is the meaning of “20in myopia”?
Author Response
Q) Why did the authors need to investigate “long-term” postoperative prediction errors? It has been demonstrated that refraction is stable 3 months after cataract surgery (Hoffer KJ, et al. Am J Ophthalmol 2015;160:403-5.e1.). If the authors wanted to investigate postoperative prediction errors after phacovitrectomy, they could evaluate prediction errors at an earlier period, not 2 years after surgery, which would led to enroll more patients.
This paper was designed to find the cause of clinical experience that myopic shifts frequently occurred after successful combined phacovitrectomy. As you said, refractive result after cataract surgery is known to show high accuracy. (line 186) However, in cases about combined phacovitrectomy, there have been many reports of gradual myopia shift unlike cataract surgery. (line 204-209) However, because these papers reported myopic shift within 12 months (line 221), we wanted to find out the tendency of myopic shift over a more extended period.
Q) although the authors mentioned that postoperative prediction errors were correlated with lens thickness (p = 0.041) and ACD (p = 0.049), and not with AL (p = 0.051), there is little difference in p values between these 3 parameters. The authors should analyze the data with larger number of patients.
Your point is correct. If we had researched with a larger number of patients, it would give more accurate results. As you said, unlike the previous report that the SRK/T formula does not cause myopic shift in the short eye, it showed a moderately significant result. In general, the shorter the axial length, the thicker the lens and the shallower the ACD (line 244). Therefore, we thought that the axial length showed moderately significant p-value by affecting the anterior chamber depth and lens thickness rather than independently affecting the myopic shift. Due to the lack of explanation on the above, we added it to the manuscript. (line 242-245, line 263)
Q) If the authors insist that the lens position in eyes after phacovitrectomy differs from that after phacoemulsification, they should provide the data of preoperative and postoperative ACD in both groups.
Thanks for pointing out. We believe that having accurate data of preoperative and postoperative ACD would be of great help to our insist. However, since the preoperative ACD is phakic ACD, and the postoperative ACD is pseudophakic ACD, these cannot be compared. Also, a method to accurately predict the IOL position before surgery has not been developed. (line 236-237) Lastly, the PCI used in this study has a limitation that reproducibility is low when measuring ACD (Annette et al. 2001, JCRS).
Q) It has been well known that the PCI AL is not influenced by retinal edema, in contrast to that measured by the ultrasound (ref. 10). Therefore, the authors should not equate papers in which ALs were measured by ultrasound to those that used PCI. In addition, they did not need to discuss the effect of macular edema on IOL power calculation by PCI.
Your point is correct. Theoretically, PCI is not affected by macular edema, because PCI measures the distance from the corneal apex to the retinal pigment epithelium layer while ultrasound measures the distance from the corneal apex to the internal limiting membrane. Nevertheless, in clinical practice, the axial length measured by PCI is shorter than actual length in patients with macular involved retinal detachment. In situations such as macular edema that result in decreased retinal transparency, PCI may show a double peak (Takashi el al, AAO, 2010). Therefore, macular edema may affect axial length in clinical practice.
Q) the authors should not discuss the relationship between AL measurement by ultrasound and refractive errors because they used PCI in this study and everyone knows that the two systems have different characteristics.
Thanks for pointing out. Several studies using Ultrasound and PCI have reported myopic shift in consistently after combined phacovitrectomy. We were wondering if this myopic shift actually happened, and we confirmed that both ultrasound and PCI result in myopic shift after combined phacovitrectomy.
Q) The authors should perform multivariable analysis to investigate the correlation between postoperative prediction error and preoperative ocular parameters because there is an interaction between preoperative parameters (for example, thicker lens leads to shallower ACD and shorter AL, as the authors mentioned).
Your point is right. If multiple independent variables were analyzed, multivariate analysis can be helpful. We did a multivariate regression analysis with continuous variable. However, we could not found any combination of variables that showed significant results. This is probably because there is a correlation between ACD, lens thickness and axial length, so there is collinearity. (ACD & lens thickness: r= -0.664, p<0.001; ACD & axial length: r=0.392, p = 0.035; lens thickness & axial length: r=-0.291, p = 0.126)
Q) Introduction is too redundant, especially 1st paragraph.
Thanks for pointing out. We deleted the redundant sentences from the 1st paragraph.
Q) P2, line78-9: Why did the authors measure ALs with not only PCI but also ultrasaound? Because, as the authors mentioned, PCI is more reliable than ultrasound.
Thanks for pointing out. As mentioned above, several studies using Ultrasound and PCI have reported myopic shift in consistently after combined phacovitrectomy. We were wondering if this myopic shift actually happened, and we confirmed that both ultrasound and PCI result in myopic shift after combined phacovitrectomy.
Q) P4, line 147: AL valued are different from those in Table 1.
Thanks for the correction. We fixed it as you said.
Q) P7, line214: What is the meaning of “20in myopia”?
Thanks for the correction. “20” is inserted incorrectly. We fixed it as you said.

Reviewer 3 Report
1) "Long-term reproducibility of axial length measurements after combined phacovitrectomy for epiretinal membrane, and refractive outcomes" This is a study of all cases up to 2 years. Consider the following title. “Two-year reproducibility of axial length measurements after combined phacovitrectomy for epiretinal membrane, and refractive outcomes”
2)The following considerations are excellent. ``A possible explanation is that vitreous and posterior lens capsule interactions disappear after vitrectomy, resulting in a shift in the IOL. In the human eye, the vitreous and lens are connected via Weigert's ligament and Cloquet's canal.Thus, when a patient is treated with combined phacovitrectomy, we propose that the procedure affects the posterior surface of the lens during wound healing and capsular fibrosis around the IOL, which may affect the position of the IOL. As such, the target refraction will be achieved by using an IOL which has a lower diopter than the calculated power.”
Author Response
Thanks for your kind review.
Q) "Long-term reproducibility of axial length measurements after combined phacovitrectomy for epiretinal membrane, and refractive outcomes" This is a study of all cases up to 2 years. Consider the following title. “Two-year reproducibility of axial length measurements after combined phacovitrectomy for epiretinal membrane, and refractive outcomes”
Thanks for the correction. We changed the title as you said “Two-year reproducibility of axial length measurements after combined phacovitrectomy for epiretinal membrane, and refractive outcomes.”
Q) The following considerations are excellent. ``A possible explanation is that vitreous and posterior lens capsule interactions disappear after vitrectomy, resulting in a shift in the IOL. In the human eye, the vitreous and lens are connected via Weigert's ligament and Cloquet's canal.Thus, when a patient is treated with combined phacovitrectomy, we propose that the procedure affects the posterior surface of the lens during wound healing and capsular fibrosis around the IOL, which may affect the position of the IOL. As such, the target refraction will be achieved by using an IOL which has a lower diopter than the calculated power.”
Thanks for your attentive comment.
Round 2
Reviewer 2 Report
Q) Why did the authors need to investigate “long-term” postoperative prediction errors? It has been demonstrated that refraction is stable 3 months after cataract surgery (Hoffer KJ, et al. Am J Ophthalmol 2015;160:403-5.e1.). If the authors wanted to investigate postoperative prediction errors after phacovitrectomy, they could evaluate prediction errors at an earlier period, not 2 years after surgery, which would led to enroll more patients.
This paper was designed to find the cause of clinical experience that myopic shifts frequently occurred after successful combined phacovitrectomy. As you said, refractive result after cataract surgery is known to show high accuracy. (line 186) However, in cases about combined phacovitrectomy, there have been many reports of gradual myopia shift unlike cataract surgery. (line 204-209) However, because these papers reported myopic shift within 12 months (line 221), we wanted to find out the tendency of myopic shift over a more extended period.
Reviewer’s comment to authors’ response: If the authors want to investigate the tendency of myopic shift over an extended period, the authors should provide the data on longitudinal change of refractive errors (e.g., at 1, 3, 6, 12, and 24 months postoperatively) and should perform statistical analyses. In the original manuscript, there was no statistically significant difference in refractive error between postoperative 1 and 2 years, which means that refractive error is stable by one year postoperatively at the latest. In addition, if the authors want to find the cause of myopic shift after phacovitrectomy, they should compare the IOL positions between in eyes with phacovitrectomy and in eyes with phacoemulsification alone.
Q) although the authors mentioned that postoperative prediction errors were correlated with lens thickness (p = 0.041) and ACD (p = 0.049), and not with AL (p = 0.051), there is little difference in p values between these 3 parameters. The authors should analyze the data with larger number of patients.
Your point is correct. If we had researched with a larger number of patients, it would give more accurate results. As you said, unlike the previous report that the SRK/T formula does not cause myopic shift in the short eye, it showed a moderately significant result. In general, the shorter the axial length, the thicker the lens and the shallower the ACD (line 244). Therefore, we thought that the axial length showed moderately significant p-value by affecting the anterior chamber depth and lens thickness rather than independently affecting the myopic shift. Due to the lack of explanation on the above, we added it to the manuscript. (line 242-245, line 263)
Reviewer’s comment to authors’ response: Again, the authors should analyze the data with larger number of patients. The authors did not address my query.
Q) If the authors insist that the lens position in eyes after phacovitrectomy differs from that after phacoemulsification, they should provide the data of preoperative and postoperative ACD in both groups.
Thanks for pointing out. We believe that having accurate data of preoperative and postoperative ACD would be of great help to our insist. However, since the preoperative ACD is phakic ACD, and the postoperative ACD is pseudophakic ACD, these cannot be compared. Also, a method to accurately predict the IOL position before surgery has not been developed. (line 236-237) Lastly, the PCI used in this study has a limitation that reproducibility is low when measuring ACD (Annette et al. 2001, JCRS).
Reviewer’s comment to authors’ response: My meaning is the authors should investigate preoperative and postoperative ACD values in both eyes with phacovitrectomy and eyes with phacoemulsification alone, and compare the values between the two groups. And, if the PCI is not appropriate to measure ACD, you should buy a machine necessary to do your research.
Q) It has been well known that the PCI AL is not influenced by retinal edema, in contrast to that measured by the ultrasound (ref. 10). Therefore, the authors should not equate papers in which ALs were measured by ultrasound to those that used PCI. In addition, they did not need to discuss the effect of macular edema on IOL power calculation by PCI.
Your point is correct. Theoretically, PCI is not affected by macular edema, because PCI measures the distance from the corneal apex to the retinal pigment epithelium layer while ultrasound measures the distance from the corneal apex to the internal limiting membrane. Nevertheless, in clinical practice, the axial length measured by PCI is shorter than actual length in patients with macular involved retinal detachment. In situations such as macular edema that result in decreased retinal transparency, PCI may show a double peak (Takashi el al, AAO, 2010). Therefore, macular edema may affect axial length in clinical practice.
Reviewer’s comment to authors’ response: But, in this study, the authors included only ERM cases, which means retina is attached in all the cases. The authors do not need to worry about the cases with detached retina.
Q) the authors should not discuss the relationship between AL measurement by ultrasound and refractive errors because they used PCI in this study and everyone knows that the two systems have different characteristics.
Thanks for pointing out. Several studies using Ultrasound and PCI have reported myopic shift in consistently after combined phacovitrectomy. We were wondering if this myopic shift actually happened, and we confirmed that both ultrasound and PCI result in myopic shift after combined phacovitrectomy.
Reviewer’s comment to authors’ response: It has been demonstrated that the PCI is superior to the ultrasound for AL measurement. Thus, you do not need to provide AL measurements by the ultrasound.
Q) The authors should perform multivariable analysis to investigate the correlation between postoperative prediction error and preoperative ocular parameters because there is an interaction between preoperative parameters (for example, thicker lens leads to shallower ACD and shorter AL, as the authors mentioned).
Your point is right. If multiple independent variables were analyzed, multivariate analysis can be helpful. We did a multivariate regression analysis with continuous variable. However, we could not found any combination of variables that showed significant results. This is probably because there is a correlation between ACD, lens thickness and axial length, so there is collinearity. (ACD & lens thickness: r= -0.664, p<0.001; ACD & axial length: r=0.392, p = 0.035; lens thickness & axial length: r=-0.291, p = 0.126)
Reviewer’s comment to authors’ response: The authors should ask a statistics expert.
The authors did not address properly any of my major concerns.